# A Multi-Physic Modelling Insight into the Differences between Microwave and Conventional Heating for the Synthesis of TiO₂ Nanoparticles

**Giulia Poppi** [1], **Elena Colombini** [1], **Diego Salvatori** [1], **Alessio Balestri** [1], **Giovanni Baldi** [2], **Cristina Leonelli** [1,*] and **Paolo Veronesi** [1,*]

1   Department of Engineering "Enzo Ferrari" (DIEF), University of Modena and Reggio Emilia, 41125 Modena, Italy; giulia.poppi@unimore.it (G.P.); elena.colombini@unimore.it (E.C.); diego.salvatori@unimore.it (D.S.); alessio.balestri@unimore.it (A.B.)
2   Ce. Ri. Col−Centro Ricerche Colorobbia Consulting, Via Pietramarina, 123, SOVIGLIANA, 50059 Vinci, Italy; baldig@colorobbia.it
*   Correspondence: cristina.leonelli@unimore.it (C.L.); paolo.veronesi@unimore.it (P.V.)

**Abstract:** Microwave-assisted synthesis of nanoparticles usually leads to a smaller and more uniformly distributed particle size compared to conventional heating (e.g., oil bath). Numerical simulation can help to obtain a better insight into the process in terms of temperature distribution or to evidence existing different temperature profiles and heating rates between the two techniques. In this paper multi-physics numerical simulation is used to investigate the continuous flow synthesis of titanium oxide nanoparticles starting from alkoxide precursors. Temperature-dependent permittivity of reactants has been measured, including the effects of permanence at the maximum synthesis temperature. A temperature homogeneity index has been defined to compare microwave and conventional heating. Results show that when using microwave heating at 2450 MHz, in the investigated conditions, a much higher temperature homogeneity of the reactants is reached. Moreover, reactants experience different heating rates, depending on their position inside the microwave applicator, while this is almost negligible in the case of conventional heating.

**Keywords:** multi-physics numerical simulation; microwave heating; nanoparticles; heating rate; TiO₂ synthesis; temperature profile; conventional heating

## 1. Introduction

Titanium dioxide (TiO₂) is a well-known and well-researched material due to its low cost, non-toxicity, chemical stability, biocompatibility, and physical, optical, and electronic properties [1]. Thanks to these features, titanium dioxide has a wide scope of applications in many different areas. As the most widely used white pigment in the world [2], it is employed in paints and varnishes [3], and glasses and ceramics, but also as a UV-blocker in sunscreens [4]. The discovery of the activity of TiO₂ nanomaterials has expanded its application range to fuel cells [5–7]; specifically, in PEMFCS applications its unique hygroscopic property facilitates water management in membranes and consequently increases conductivity. In dye-sensitized solar cells [8–10] the photoanode can be fabricated using doped titania particles with the increased ultraviolet–visible (UV–vis) absorbance and the reduced band gap of 2.8 from 3.25 eV. The chemi-resistive behavior of nano-anatase thin films exposed to oxidizing and reducing gases (O₂, H₂, and ethanol) in the temperature range between 300 and 400 °C allowed the development of gas sensors [11–15]. Additional applications has been found in the fields of capacitors [16–20], photocatalytic corrosion-protective coatings [21–27], and thin films transistors [28–31]. Titanium dioxide photocatalysis has also proven to be an effective method of purifying contaminated air and water under UV–visible irradiation [32–37]. The photocatalytic effect of TiO₂ nanomaterials can be efficiently exploited due to their high surface-volume ratio, which offers an

increasing light absorption rate. Doping techniques have also been employed to broaden the effective range of light sensitivity from the UV to the visible light region of the spectrum [38]. Among the possible processing routes to fabricate such modified TiO$_2$, sol–gel synthesis provides nanoparticles already suspended in water, ready for further processing. TiO$_2$ nanoparticles are usually synthetized via an acid-catalyzed hydrolysis of titanium (IV) alkoxide followed by condensation [1,4]. Such a synthetic route is at the basis of a patented commercially available product and will be investigated in this paper [39]. Sol–gel synthesis has the advantages of a low cost, low operating temperature, and high chemical homogeneity and purity [40]. However, it produces low crystallinity products [40,41], which need to be improved by further thermal treatment causing an increase in the particle size [42]. In addition, sol–gel processing requires long reaction times and it may also be energy-consuming [43].

The aim of this work is the use of numerical simulation to gain a better insight into the microwave heating of alkoxide precursors, in an aqueous solution, to synthetize titanium dioxide nanoparticles. This can lead to a further process intensification of the synthetic route, using microwave heating to speed up the preliminary phases of the sol–gel process. Moreover, the proper use of microwave heating is expected to lead to the synthesis of smaller nanoparticles having a narrower particle size distribution. Numerical simulation is used also to maximize the energy efficiency of the microwave applicator, by minimizing the power that is reflected back to the microwave source, hence maximizing the energy converted into heat inside the load.

As a matter of fact, microwave heating is based upon the ability of materials to absorb and transform electromagnetic energy into heat [44]. In the case of a polar solvent, such as water and alcohols, the electromagnetic wave interacts with the material leading to a rapid and homogeneous volumetric heating, which is a direct consequence of dielectric losses caused by dipolar polarization [44,45]. Microwave-assisted sol–gel synthesis is a relatively novel method to produce TiO$_2$ nanoparticles, but it has sparked a remarkable interest due to its numerous advantages [46], such as a reduction in processing time and temperature, which provide a decrease in costs and energy consumption [40,45], increase in the kinetics of crystallization [45], and eco-friendliness [47]. Other reported advantages are phase purity, better reliability and reproducibility [48], homogeneous heating [43], and a reduction in the particle growth during the process [40].

In this framework, a recently EU-funded project, SIMPLIFY (Sonication and Microwave Processing of Material Feedstocks) [49], aims to bring technical innovation in the processing of many kinds of materials according to the process intensification principles, especially by the use of ultrasound and microwaves as effective tools for energy supply. One of the three main research lines was focused on the TiO$_2$ nanoparticle syntheses by microwaves and ultrasound in a PFR (Plug Flow Reactor). Within this line, a computer model was built to simulate and investigate the steady state temperature distribution occurring in the designed microwave applicator. A comparison between conventional (oil bath) and microwave heating is presented in this study, in order to assess possible temperature distribution differences between the two heating methods and, hence, explain the experimentally observed higher particle size uniformity occurring when using microwave processing.

## 2. Materials and Methods

Hydro alcoholic suspensions of TiO$_2$ nanoparticles can be obtained using Ti(OiPr)$_4$ (titanium isopropoxide, technical grade) and a premixed aqueous HCl solution (technical grade) with a small amount of a surfactant, commercially known as Triton X-100 (Sigma Aldrich, Merk Life Science S.r.l., Milan, Italy, laboratory grade). The amounts of three such components present in the mixture are based on a known procedure used to obtain a commercially available product [39], namely: 32% HCl = 3.1% Bi-distilled water = 96.9% Triton X-100 = 0.013%, reacting as follows

$$Ti(OiPr)_4 + 2\,H_2O \xrightarrow{HCl} 4\,iPrOH + TiO_2$$

The electric field orientation depends on time with a frequency of 2.45 GHz (the electric field vector switches its orientation approximately every $10^{-12}$ s). In the case of polar molecules, the electric field induces rotations of such molecules, but introduces a delay with respect to the exciting field, which is related to the nature of the molecules and their bonds. This delay between electromagnetic stimulation and molecular response is the physical origin of the dielectric loss, responsible for fast and volumetric microwave heating. Because the process is not dependent upon heat transfer, such as heat conduction, the result is a rapid and localized heating, depending on electric field strength, targeting dipoles, or existing ionic species. Both phenomena are accounted for in the measurement of the effective permittivity of dielectrics.

In order to predict the response of the reactants during MW irradiation, we proceeded with the experimental measurements of the permittivity of the reactants' mixtures and their evolution into products. An Agilent 85070E Dielectric Probe Kit (Agilent Technologies, Santa Clara, CA, USA) operating in the frequency range 1–3 GHz was used under conventional heating (hot plate) and monitoring the temperature by an optical probe (Neoptix T1 Optic Temperature Probe, Neoptix, Quebec city, Canada) in the range 25–80 °C. A first-degree interpolating function of the measured properties was derived and used to describe load changes as a function of the reaction temperature

$$\varepsilon'(T) = -0.06 \cdot T + 72.589 \tag{1}$$

$$\varepsilon''(T) = 0.15 \cdot T - 22.975 \tag{2}$$

Due to the relevant changes in permittivity occurring as the reaction proceeds, a twin microwave applicator geometry was devised, having the first applicator dedicated to the heating stage (with reference to the permittivity values of Figure 1a and Equations (1) and (2)) and the second to the holding stage (Figure 1b).

A coil-like arrangement ("helix") of PTFE pipes was selected to achieve a controllable continuous flow of reactants inside the microwave applicator. The diameter of the PTFE pipes was chosen taking into account the calculated microwave penetration depth, so that each portion of the load could be directly exposed to microwaves. Considering also the temperature dependence of the permittivity, a diameter of 15 mm was chosen, being of the same order of magnitude of the calculated microwave power penetration depth at 2.45 GHz.

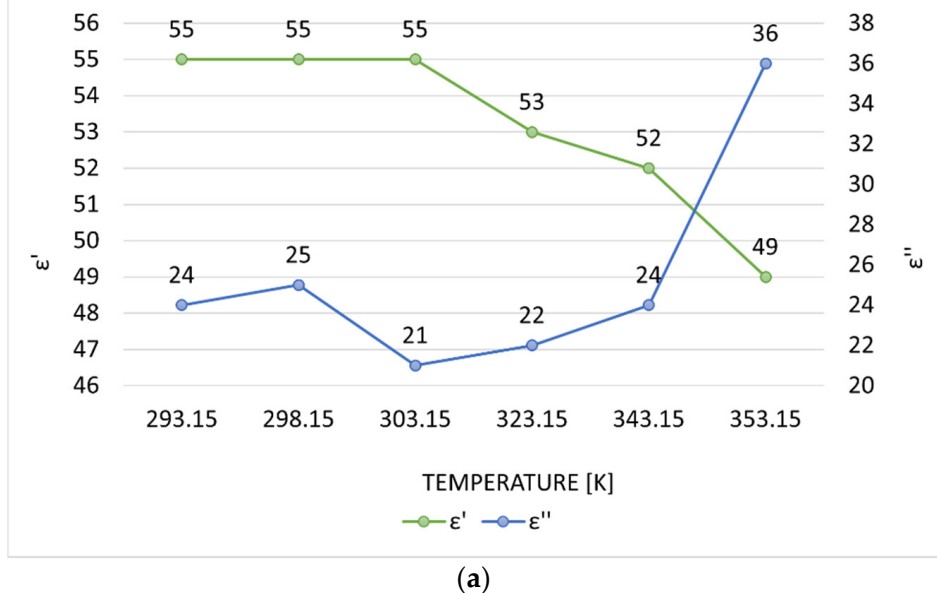

(**a**)

**Figure 1.** *Cont.*

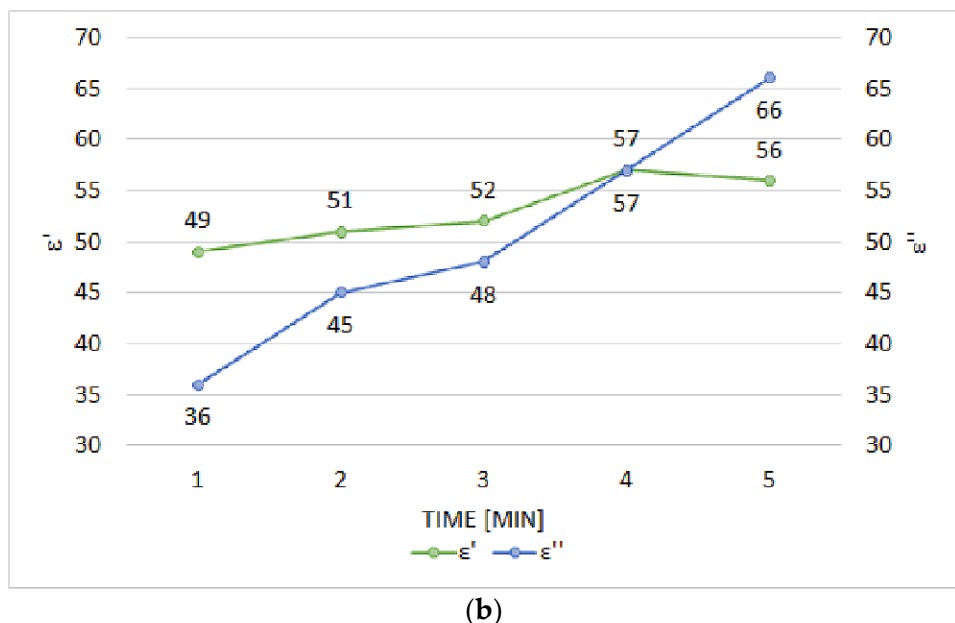

**(b)**

**Figure 1.** Permittivity of the load at 2.45 GHz: (**a**) as a function of temperature; (**b**) as a function of the holding time at 353.15 K (80 °C).

Figure 2 shows the geometry of the proposed microwave applicator, namely a hexagonal prismatic reactor powered on one side by a WR340 waveguide and containing the coil-like arrangement of PTFE pipes with the reactants. The prismatic geometry of the applicator was selected because, according to previous studies [50], it allows to loads to be processed in the region of predominant magnetic or electric fields. A preliminary optimization of the applicator's geometry in terms of inscribed circumference radius and load position was conducted using the commercial software QuickWave 3D, focusing on energy efficiency (minimization of the reflection coefficient).

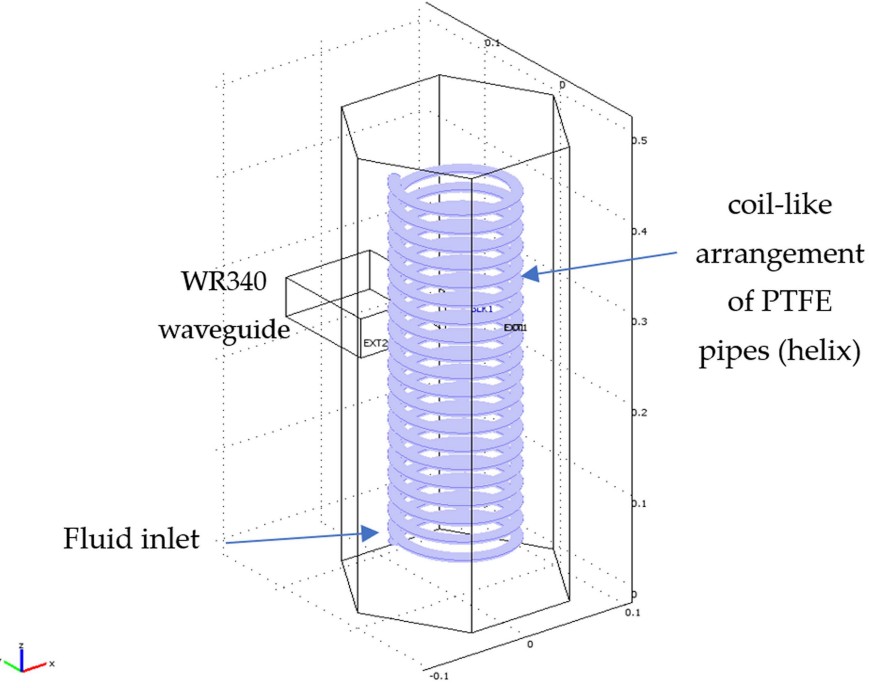

**Figure 2.** Geometry of the heating stage chamber of the microwave applicator.

In order to assess the 3D temperature distribution inside the load and to take into account the flowing fluid, multi-physics numerical simulation was conducted using the commercial FEM-Finite Element Method software COMSOL Multiphysics® version 3.5a. The model presents a two-way coupling (electromagnetic heating and non-isothermal flow) among three different physics, namely electromagnetic waves, heat transfer, and fluid flow. The material properties are temperature dependent and, consequently, affect the electromagnetic field distribution in the microwave applicator. Microwave heating simulation is performed at 2.45 GHz with an average fluid speed of 0.5 m/s at the inlet section. We assumed laminar flow in the helix-shaped pipe conveying the liquid load. The hypothesis of outward heat flux by natural convection of air within the applicator has been used as a boundary condition on the inner pipe walls for the microwave heating. The microwave applicator's walls have been modeled as PEC (Perfect Electric Conductor) and the waveguide that feeds the reactor has been excited in the fundamental mode $TE_{10}$, introducing an input power of 8 kW into the applicator.

The model simulating conventional heating presents a one-way coupling (non-isothermal flow) between heat transfer and fluid flow physics. The same properties and conditions used in the case of microwave heating were applied for the liquid load, except for a thermal boundary condition set on the pipe inner walls, i.e., a constant temperature of 90 °C. This is used to simulate the helix immersed in an oil bath held at such constant temperature.

Both models were solved in transient and steady state conditions, in order to assess, respectively, the temperature evolution of the load in time during the early stages of heating, as well as during the continuous flow processing of the load.

In order to better evaluate possible differences in temperature distribution deriving from the heating techniques simulated, a temperature homogeneity index has been defined as the ratio between the average temperature of the fluid and the standard deviation of the temperature values in the whole volume of the load. Such an index assumes high values in the case of homogenous temperature distribution in the load and if the average temperature is high.

## 3. Results and Discussion

### 3.1. Preliminary Applicator Optimization

The dimensions of the hexagonal applicator have been optimized in terms of the applicator equivalent radius (R) for a given helix-shaped load configuration. The $|S_{11}|$ parameter was used for this purpose. $|S_{11}|$ is the reflection coefficient defined as the ratio between the electric field leaving the input port (1) and the electric field entering the input port (1), under the condition that no signal enters the output port [51]. Consequently, the optimal dimensions of the applicator are those for which the reflection coefficient assumes the minimum value at the operating frequency of the microwave source, which is 2.45 GHz. This means that, during the process, the amount of microwave energy reflected back to the generator is minimum, the remaining quota being absorbed and converted into heat by the load.

Therefore, $|S_{11}|$ at 2.45 GHz has been calculated as R varies in the range 90–120 mm. The results are plotted in Figure 3 and show that the minimum value of $|S_{11}|$ at 2.45 GHz has been obtained for a radius R = 108.8 mm.

All the results presented in the next sections refer to this optimized geometry of the hexagonal applicator. In the optimized conditions, moreover, the variation in the reflection coefficient with frequency presents a rather large band minimum, which is favorable when using microwave sources such as magnetrons, which emit microwaves over a wider band, and are not necessarily centered at 2.45 GHz (see supplementary material, Figure S1, where numerical simulation data are accompanied by modeling validation, performed by measurement on the built applicator using an Agilent HP8753D Vector Network Analyzer).

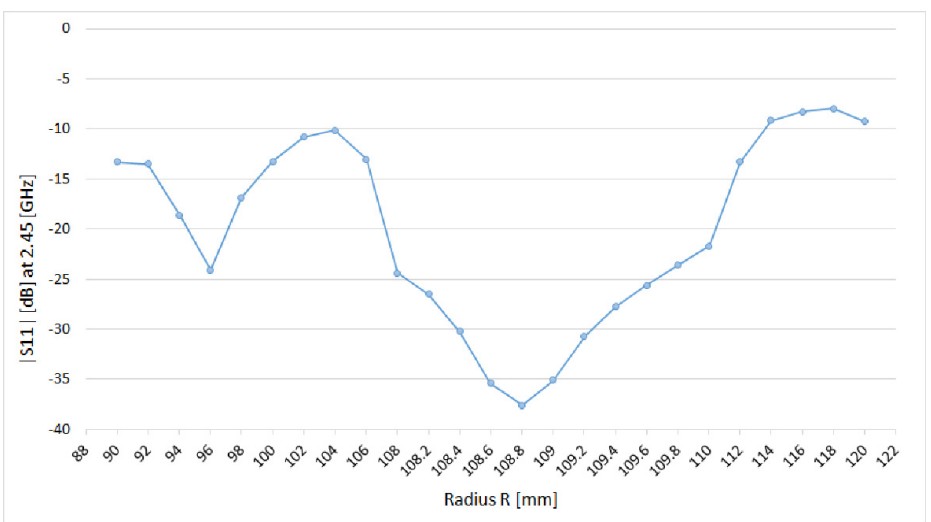

**Figure 3.** Variation in |S11| parameter (dB) calculated at the operating frequency of 2.45 (GHz) as function of the radius R of the circumscribed circle of the hexagonal applicator.

*3.2. Comparison between Conventional and Microwave Heating*

The simulation of both conventional and microwave heating in transient conditions was used to assess the temperature distribution of the flowing fluid in the early stages of the heating, and to determine when the process reaches the steady state conditions. In this framework, numerical simulation is a very powerful tool, as it allows investigating the temperature distribution inside each portion of the load. This is useful to gather a deeper understanding of possible different heating profiles existing while processing the fluid. Such information would not be easily accessible by direct temperature measurements on the load volume, as the insertion of probes would result in a perturbation of the electric or temperature field.

Figure 4 shows the simulated temperature distribution in a cross section of the load (Figure 4a indicates the position of the cross section with respect to the microwave inlet) along the height of the helix, at a distance from the applicator axis equal to the helix radius. The reported values on the *x*-axis indicate the height relative to the bottom of the applicator, where the fluid inlet is positioned.

The temperature plots of Figure 4 indicate, for each position on the x-axis, how the temperature increases with time in the cross section of the PTFE pipe where the load is contained. For a given 15 mm interval on the x-axis, corresponding to one helix turn, and time, it is possible to visualize the temperature distribution existing in such a section of the load.

In conventional heating (Figure 4b), the walls of the helix are at a constant temperature of 363.15 K throughout the process, as a consequence of the boundary thermal conditions set. This results in all temperature profiles of each helix turn starting and ending at that temperature. As the fluid flows into the helix, starting from 293.15 K, the temperature of the fluid in the first helix turns remains approximately at that temperature, except for the thin layers of fluid near to the walls, where heat exchange with the pipe walls occurs. The heating on each section of the pipe is highly non-homogeneous, especially in the early stages of heating, with the inner part significantly colder than the outer part, as shown by the accentuated curvature of the graph lines at all times. This expected behaviour is due to the assumption of laminar flow in the pipes, which simulates the real processing conditions. The temperature then gradually increases as the fluid moves towards the outlet section (top of the applicator) and the heating becomes more homogeneous in each cross section of the helix. The steady state is reached after 30 s, and in these conditions, the fluid in the upper half of the helix (outlet section) reaches the desired target minimum temperature of 353 K (almost 80 °C).

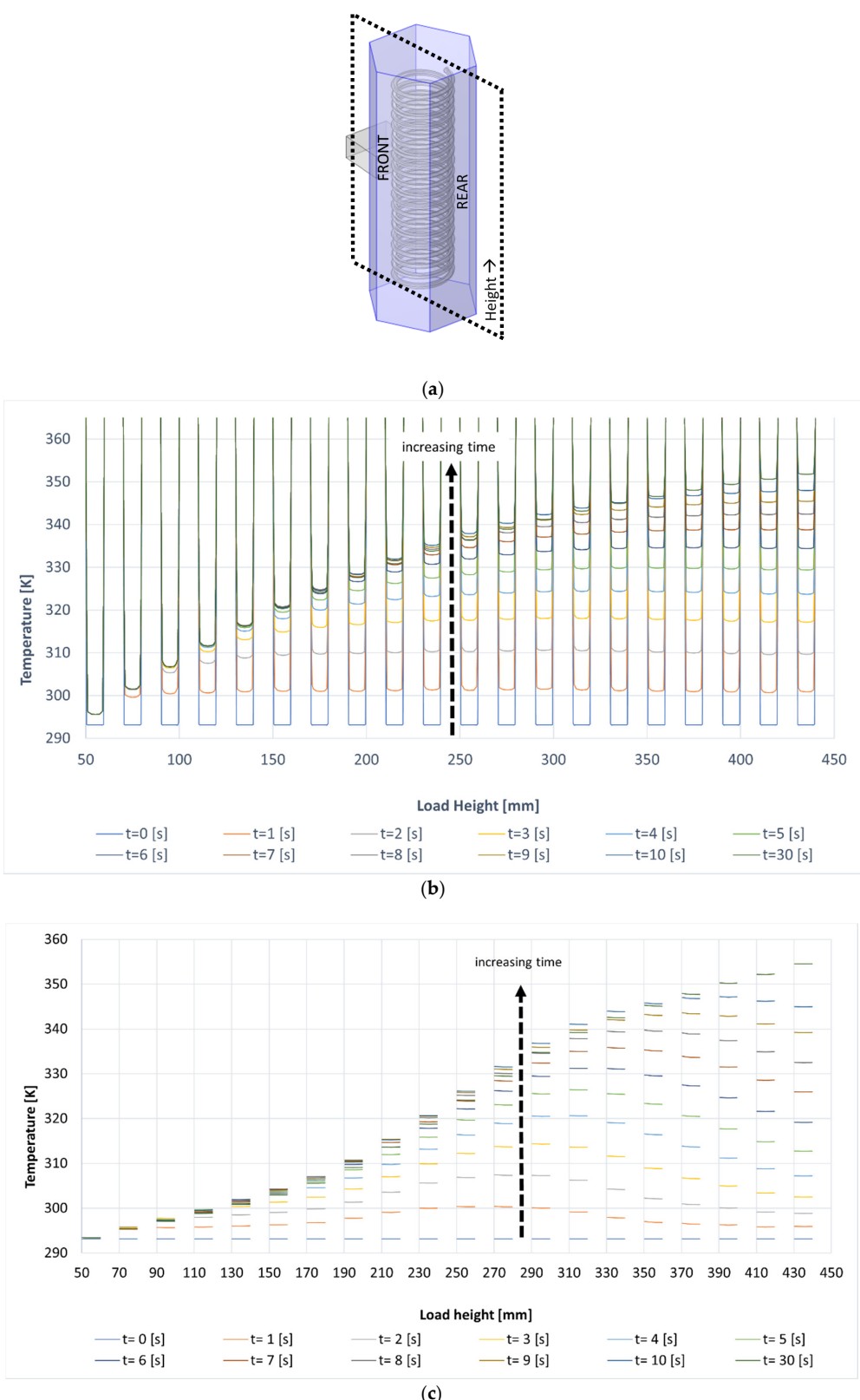

**Figure 4.** (**a**) Position of the cross section used to extract the data and temperature distribution in the load section along the height of the helix in transient conditions: (**b**) conventional heating and (**c**) microwave heating. Dotted arrows indicate the direction of increasing simulated heating time, from 0 to 30 s.

In the case of microwave heating (Figure 4c), the continuous lines show the fluid warming up from the centre of the pipe. This is typical of the volumetric heating occurring within the microwave penetration depth and with thermal boundary conditions of "cold walls". In fact, the PTFE pipe walls at the beginning are at room temperature and they are practically not directly heated by microwaves. This, in turn, generates this inversion of the temperature profiles, with respect to conventional heating (hot walls). The graph also indicates that the heating rate of a given helix turn is quite different depending on its distance (along the vertical axis) from the feeding waveguide, and this will be clearer in steady state conditions, as discussed later.

Steady state conditions are reached after 30 s, and the temperature at the outlet section slightly overcomes the target temperature value, reaching 358.05 K (almost 85 °C). Compared to conventional heating, in the case of microwave heating, a more homogeneous heating occurs, as shown by the flatter curvature of the temperature plots of the fluid in each one of the helix revolutions, at all times.

Figure 5 shows the temperature distribution of the load along the helix in steady state conditions, in the case of conventional heating (Figure 5a) and microwave heating (Figure 5b,c), respectively. In Figure 5a, the slice plot shows the temperature profile existing in different cross sections of the pipes. The slice plots are obtained using five vertical cross sections, passing through the helix-like arrangement of the PTFE pipes. Hence, each ellipse in the slice plot gives an indication of the temperature distribution existing in that particular region, at a given time. Results provide a further confirmation of the existing temperature gradients from the outer to the inner part of the pipe. This is particularly evident near the bottom of the load, where the cold reactants enter the heated applicator.

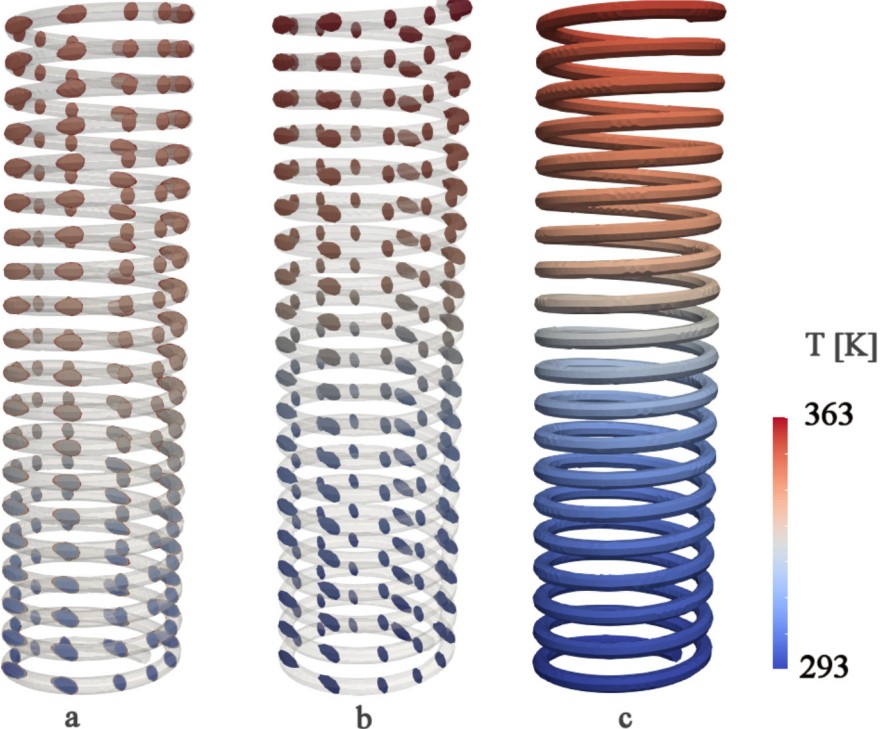

**Figure 5.** Temperature distribution of the load in steady state conditions: (**a**) slice plot, conventional heating; (**b**) slice plot, microwave heating, (**c**) surface plot, microwave heating.

Figure 5b shows a much more homogenous temperature distribution existing in all cross sections of the slice plot, as expected from the plots of Figure 4. Moreover, the volume plot of Figure 5c evidences that most of the temperature rise of the load occurs at the mid height of the helix, indicating the existence of different heating rates.

For this reason, a further plot was derived (Figure 6), showing the heating rates experienced by the load along the applicator height. They are calculated as the difference in average temperature between two neighboring turns of the helix-shaped load. Despite being temperature gradients, for a constant speed of the fluid, they can be assumed to be an indication of how rapidly temperature is increased and, in practice, as heating rates.

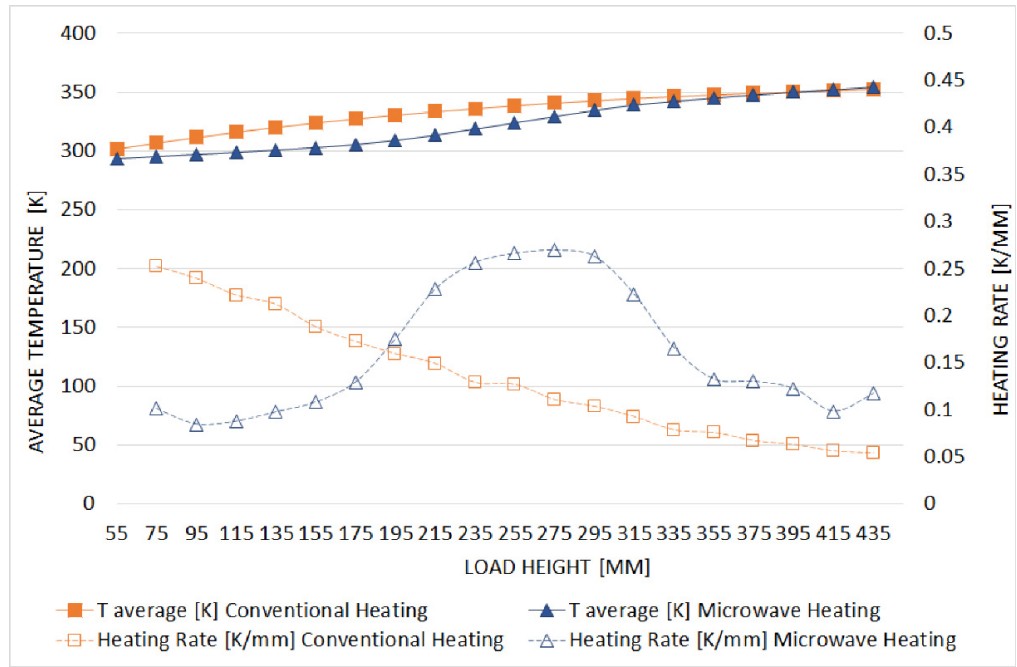

**Figure 6.** Average temperatures and heating rates (dotted lines) of the load along the applicator height in the front of the helix with respect to the waveguide, in the case of conventional and microwave heating.

Results in Figure 6 show that in the case of microwave heating, the way the load reaches the target average temperature at the outlet section is completely different from what happens in conventional heating, with most of the temperature rise occurring when the load is near to the microwave input port (waveguide). This is particularly evident when considering the heating rate, which has an almost constant trend as heating proceeds, in the case of conventional heating, but it has a pronounced peak in the central regions of the applicator in the case of microwave heating. This behaviour is experienced also in the portions of the load positioned on the opposite side with respect to the microwave waveguide input (see supplementary material, Figure S2), but to a minor extent due to the higher electric field strength in such regions, which is due to the applicator geometry (see supplementary material, Figure S3).

In order to better quantify the differences between the two heating modes of the continuous process of $TiO_2$ synthesis, a temperature homogeneity index has been defined as

$$\text{Temperature Homogeneity Index (THI)} = \frac{T_{av}}{\sigma} \tag{3}$$

where $\sigma = \frac{\sqrt{(T-T_{av})^2}}{N}$ is the variance, N the number of mesh elements of the helix, and $T_{av}$ the average temperature of the fluid in a reference volume or cross section. Both the variance and the average temperature of the fluid were calculated in steady state conditions, in each revolution of the helix, for the cross sections indicated in Figure 4a. A high value of the homogeneity index is desirable, as it indicates that heating occurs rapidly and with a small variance in temperature distribution in the cross section of the load.

The obtained results are shown in Figure 7, for conventional heating (Figure 7a) and microwave heating (Figure 7b).

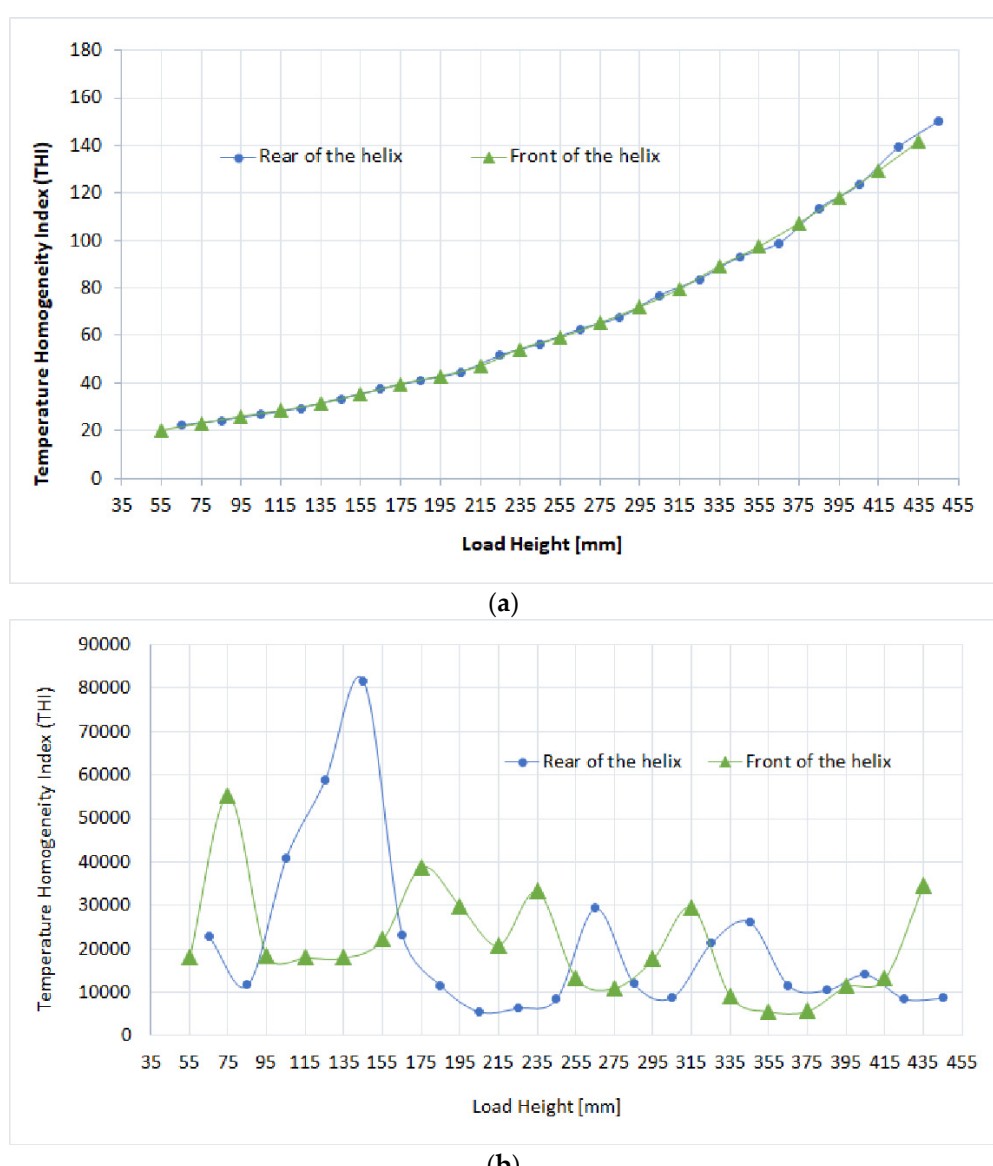

**Figure 7.** Temperature homogeneity index (THI) as function of the height of the load inside the helix: (**a**) conventional heating; (**b**) microwave heating.

Comparing the values of the temperature homogeneity indexes of the two graphs of Figure 7, it is evident that in the simulated conditions, microwave heating involves a much more uniform heating of the load compared to conventional heating. In fact, the THI of microwave heating is larger than that of conventional heating by two orders of magnitude. This reflects what is expected when using oil baths for heating and laminar flow, which result in a stratification of temperatures from the center to the outer regions of the circular cross section of the pipe, while heat transfer occurs. Instead, the volumetric generation of heat offered by microwave heating helps to reduce such phenomena. However, it should be pointed out that in the case of conventional heating there is practically no effect of the THI in two neighboring turns of the helix (indicated as "front" and "rear"), while in the case of microwave heating there is a strong influence of the position of the helix turn with respect to the microwave inlet waveguide. This can be proficiently exploited to achieve a better control over heating rates and temperature homogeneity, by positioning multiple microwave sources on the applicator walls.

## 4. Conclusions

Numerical simulation results demonstrate that during microwave heating, a much higher temperature homogeneity and spatially different heating rates can be achieved, compared to conventional heating by oil bath. These different heating features could explain the experimentally observed differences in the field of microwave-assisted nanoparticle synthesis, where a better control of particle average size is encountered [52]. As demonstrated by the authors in a previous paper, addressing a completely different microwave applicator and load geometry [53], the narrower temperature distribution in microwave heating can correspond to the fact that a higher portion of the reactant volume is in the conditions of the maximum nucleation rate and minimum growth rate. On the contrary, a wider temperature distribution would lead to portions of the load volume outside this condition; hence, to a possible more pronounced growth or less nucleation. In this framework, the narrower temperature distribution offered by microwave heating would mean that many small particles are formed. Conversely, the broader temperature distribution by conventional heating indicates that larger portions of the load can also undergo substantial growth, i.e., particles are nucleated, and some of them progressively grow in size, leading to a wider particle size distribution in comparison with the previous case. This is in agreement with the known conditions required to prepare highly uniform nanoparticles: it is necessary to induce a short burst of nucleation temporally separated from the subsequent growth stage [54]. Moreover, it is worth remembering that a narrow energy distribution is one of the main requirements for process intensification, as reported by Van Gerven and Stankiewicz [55].

However, it should be noticed that the obtained temperature distributions are dependent upon the type of reactor, load position and shape, and the nature (permittivity, thermal properties) of the reactants, and this could explain the poor reproducibility of results in studies conducted using different reactors or conditions. For instance, also in this study, the results slightly change when addressing different positions of the load ("front" and "rear" position). Moreover, the model itself introduces some simplifications, and in particular, the condition of laminar flow could not be met throughout the whole duration of the synthesis process because the precursor is subjected to changes in viscosity and volume.

Last but not least, the temperature homogeneity index-THI trend in the case of microwave heating varies significantly depending on the measurement position: facing the waveguide inlet, or on the opposite side. Despite a THI much higher than in the case of conventional heating, in the position far from the waveguide, the temperature distribution in the cross section of the load is less homogeneous compared to the front, as illustrated by the lower THI values in Figure 7b. This is due not only to the higher electric field strength in the regions of the applicator where the waveguide input is positioned, but also to the fact that the temperature outside the helix is set at room temperature (microwaves do not heat the surrounding environment). This results in the cooling down of the outer regions of the fluid. This cooling effect is rather compensated by the pronounced heat generation occurring in the regions of higher electric field strength, but it becomes not negligible in regions of lower electric field strength, such as the bottom and top of the applicator, and the sides positioned opposite to the waveguide inlet.

Such results suggest that a further improved design to improve microwave heating homogeneity, should include waveguides positioned on each wall of the applicator, and not only on one side.

**Supplementary Materials:** The following are available online at https://www.mdpi.com/article/10.3390/pr10040697/s1, Figure S1: Variation in the simulated reflection coefficient |S11| (dB) with frequency (Hz) in the optimized conditions (inner diameter of the hexagonal applicator: radius = 108.8 mm) and measured reflection coefficient; Figure S2: Average temperatures and heating rates of the load along the applicator height in the rear of the helix with respect to the waveguide, in the case of conventional and microwave heating, Figure S3: Electric field strength in the applicator, slice plot, showing the higher electric field strength in the central regions of the applicator, Table S1: |S11| parameter (dB) values calculated at the operating frequency of 2.45 (GHz) as a function of the diameter (mm) of the inner circumference of the hexagonal applicator; Table S2: Temperature

Homogeneity Index (THI) as function of the height of the load inside the helix: conventional heating and microwave heating, front of the helix with respect to the waveguide; Table S3: Temperature Homogeneity Index (THI) as function of the height of the load inside the helix: conventional heating and microwave heating, rear of the helix with respect to the waveguide; Table S4: Values of average temperatures and heating rates of the load along the applicator height in the front of the helix with respect to the waveguide, in the case of conventional and microwave heating; Table S5: Values of average temperatures and heating rates of the load along the applicator height in the rear of the helix with respect to the waveguide, in the case of conventional and microwave heating.

**Author Contributions:** Conceptualization, P.V. and C.L.; methodology, P.V., E.C. and G.P.; software, G.P. and E.C.; validation: G.B., D.S. and A.B.; data curation, G.P.; writing—original draft preparation, G.P.; writing—review and editing, P.V., G.B. and C.L.; funding acquisition, C.L. and G.B. All authors have read and agreed to the published version of the manuscript.

**Funding:** This work was carried out in the context of a larger European project named: "Sonication and Microwave Processing of Material Feedstock (SIMPLIFY)" supported by the European Union's Horizon 2020 research and innovation program under Grant Agreement No 820716.

**Institutional Review Board Statement:** Not applicable

**Informed Consent Statement:** Not applicable

**Data Availability Statement:** The data presented in this study are available on request from the corresponding author.

**Conflicts of Interest:** The authors declare no conflict of interest.

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
