# Peer review of "A Multi-Physic Modelling Insight into the Differences between Microwave and Conventional Heating for the Synthesis of TiO2 Nanoparticles"

_processes, doi:10.3390/pr10040697_

Round 1
Reviewer 1 Report
The submitted article investigates the applicability of microwaves in the chemical synthesis of TiO2 nanoparticles and compares the MW-heating technique to conventional heating by using multiphysic approach. The topic and the subject is relevant and imporant as well, since TiO2 is a widely and commonly used compound in many different fields of science and has numerous applications.
My questions and remarks:
- The title in the submission system is not the same as the title in the uploaded manuscript.
- Although 49 different references are cited in the Introduction, they are usually stacked/cumulated, and therefore the section itself is quite short. The applicability of TiO2 should be explained in more details (ie. some of these references should be "expanded"), as well as the molecular background of MW-assisted chemical synthesys.
- The unit of measurement for temperature in the text is sometimes °C and sometimes K, and though it is not particularly hard to convert them (even via head counting), it would definitely improve the flow of the text if either sticking to one unit of measurement, or provide its alternative for example in brackets.
- In Figure 2b, Figure 3, and some other cases the text color of the x and y-axis is a bit faint, using black (like in Figure 2a) would be better
- In Line 121 it is mentioned that the optimization was done focusing on energy efficiency. Please give more details about how this exact model provide optimal conditions for energy transfer and distribution. Also, during either Discussion or Conclusions, point out more clearly how dielectric (MW) heating is more efficient in this particular case, ie. in the synthesis of TiO2.
- In Figure 4b and 4c the x-axis does not have the same spacing of distribution, which makes the comparison harder.
- What is the exact purpose of the dotted arrow named "increasing time" in the middle part of these (Fig 4b, 4c) diagram (since it is not at the same spot in 4b and 4c)
- In Figure 6, the data-markers for the avg. T and the Heating rate is really indistinguishable (although it is not so hard to figure out which is which), so using a different, more obviously different one is advised
- I suggest mentioning the limitations of the multiphysic stimulation in this particular case in a bit more detailful way (either in the Discussion part or in the Conclusions), or pointing out what differences could happen and why in case of a real MW-assisted synthesis during these conditions and setup
Author Response
Thank you for your comments and suggestions. They have indeed helped us in the amelioration and intelligibility of figures of the manuscript.
Detailed answers to your comments are attached in the pdf file

Reviewer 2 Report
Good job on the numerical simulation of the temperature distributions and heating rates during microwave/conventional heating synthesis of TiO2 nanoparticles. However, the authors should pay some more attention on the microwave field intensity distribution inside the microwave cavity that gives direct effect on the heating results.
Reviewer 3 Report
This paper reported that the performance of their reactor based on the FEM simulation. I think there is almost no description of science of TiO2 nanoparticles. The title should be changed to represent the contents of the present paper. Introduction should also be revised based on the contents of the paper.
Specific comments.
Line 73: I think "," is not necessary.
In section 2. I think that the first two paragraphs are not needed.
Line 128: What is "FEM"?
Around lines 219: Please define the precision of temperature.
Figure 5. It is difficult to understand slice plot for materials scientists. more explanation are required.
Figure 6. It is very difficult to recognize values of x-axis. Please revise figure 6. It is also difficult to distinguish symbols of T and Heating Rate in the figure.
Figure 7. It is also difficult to recognize the vales on x-axis in these figures.
Round 2
Reviewer 1 Report
Dear Authors,
I'd like to thank you very much for your prompt and detailed response to my remarks, questions and suggestions; I hope they have served the purpose of making your remarkable work even better. I have carefully read and reviewed the revised version of your manuscript, and now I'm more than positive to state that it is sufficiently improved for acceptance and publishing. I would also like to congratulate on your outstanding and important work in the research field you are working with.